# Novel Prediction Score for Arterial–Esophageal Fistula in Patients with Esophageal Cancer Bleeding: A Multicenter Study

**DOI:** 10.3390/cancers16040804

**Published:** 2024-02-16

**Authors:** Sz-Wei Lu, Kuang-Yu Niu, Chu-Pin Pai, Shih-Hua Lin, Chen-Bin Chen, Yu-Tai Lo, Yi-Chih Lee, Chen-June Seak, Chieh-Ching Yen

**Affiliations:** 1Department of Emergency Medicine, Keelung Chang Gung Memorial Hospital, Keelung 204, Taiwan; lala10606@gmail.com; 2Department of Emergency Medicine, Tri-Service General Hospital SongShan Branch, National Defense Medical Center, Taipei 105, Taiwan; 3Department of Emergency Medicine, Chang Gung Memorial Hospital, Linkou Branch, Taoyuan 333, Taiwan; peidra.niu@gmail.com (K.-Y.N.); jessica929lee@gmail.com (Y.-C.L.); 4Division of Thoracic Surgery, Department of Surgery, Lotung Poh-Ai Hospital, Yilan 265, Taiwan; bcbjohnny@gmail.com; 5Department of Gastroenterology and Hepatology, New Taipei Municipal Tucheng Hospital, New Taipei City 236, Taiwan; jeffrey3436@hotmail.com; 6Department of Emergency Medicine, New Taipei Municipal Tucheng Hospital, New Taipei City 236, Taiwan; ans76ers@gmail.com (C.-B.C.); julianseak@hotmail.com (C.-J.S.); 7Chang Gung Memorial Hospital, Linkou Branch, Taoyuan 333, Taiwan; jaxon3579@gmail.com; 8College of Medicine, Chang Gung University, Taoyuan 333, Taiwan; 9Institute of Emergency and Critical Care Medicine, National Yang Ming Chiao Tung University, Taipei 112, Taiwan

**Keywords:** esophageal cancer, cancer bleeding, gastrointestinal bleeding, arterial–esophageal fistula, predictive score

## Abstract

**Simple Summary:**

In patients with esophageal cancer bleeding, the presence of tumor ulcer and arterial–esophageal fistula (AEF) is a common occurrence. Notably, AEF is associated with an exceptionally poor prognosis, yet there is no prediction score to estimate its occurrence rate. Therefore, we introduce a novel model, the HEARTS-Score, for predicting AEF in esophageal cancer bleeding patients. This predictive model effectively distinguishes patients at risk, as evidenced by a *c-statistic* of 0.82 (95% CI 0.72–0.92). By employing this model, clinicians can more objectively differentiate between high-risk and low-risk patients, facilitating more efficient clinical decision-making, diagnostic planning, and subsequent treatment strategies.

**Abstract:**

Purpose: To develop and internally validate a novel prediction score to predict the occurrence of arterial–esophageal fistula (AEF) in esophageal cancer bleeding. Methods: This retrospective cohort study enrolled patients with esophageal cancer bleeding in the emergency department. The primary outcome was the diagnosis of AEF. The patients were randomly divided into a derivation group and a validation group. In the derivation stage, a predictive model was developed using logistic regression analysis. Subsequently, internal validation of the model was conducted in the validation cohort during the validation stage to assess its discrimination ability. Results: A total of 257 patients were enrolled in this study. All participants were randomized to a derivation cohort (*n* = 155) and a validation cohort (*n* = 102). AEF occurred in 22 patients (14.2%) in the derivation group and 14 patients (13.7%) in the validation group. A predictive model (HEARTS-Score) comprising five variables (hematemesis, active bleeding, serum creatinine level >1.2 mg/dL, prothrombin time >13 s, and previous stent implantation) was established. The HEARTS-Score demonstrated a high discriminative ability in both the derivation and validation cohorts, with c-statistics of 0.90 (95% CI 0.82–0.98) and 0.82 (95% CI 0.72–0.92), respectively. Conclusions: By employing this novel prediction score, clinicians can make more objective risk assessments, optimizing diagnostic strategies and tailoring treatment approaches.

## 1. Introduction

Esophageal cancer is characterized by its aggressive nature and poor overall survival rates. It ranks as the ninth most common cancer globally and stands as the sixth leading cause of cancer-related mortality [1,2]. Various risk factors contribute to its development, including tobacco smoking, alcohol consumption, lower socioeconomic status, suboptimal oral hygiene, the presence of Barrett’s esophagus, and obesity [3]. Esophageal cancer exhibits substantial geographical variability in its incidence. In Taiwan, like many other Asian nations, the predominant histological subtype of esophageal cancer is squamous cell carcinoma. The rising incidence of esophageal cancer in Taiwan has emerged as a significant and pressing public health concern. This prevalence may be linked to dietary practices and specific cultural habits, such as betel nut chewing [4,5].

Patients with esophageal cancer present to the emergency department (ED) for a variety of reasons, including dysphagia, vomiting, diarrhea, abdominal pain, malnutrition, infections, and even severe conditions like cancer bleeding. Previous research indicates that approximately 5.7% of ED visits in patients with esophageal cancer are attributed to upper gastrointestinal bleeding (UGIB). When investigating the causes of UGIB, it has been found that around 53.8% of patients with esophageal cancer who experience UGIB do so due to tumor ulcer. Moreover, approximately 12% of these cases develop a life-threatening condition known as arterial–esophageal fistula (AEF). This emphasizes that AEF is not uncommon in this specific patient population [6,7]. AEF itself carries an exceptionally high mortality rate, ranging from 46.6% to 63% [7,8,9,10], and typically necessitates emergency surgical intervention or the use of vascular or esophageal stents to temporarily control bleeding. Delayed diagnosis is associated with a poorer prognosis, emphasizing the critical importance of prompt detection and treatment. However, due to its episodic and nonspecific symptoms, AEF is often initially misdiagnosed as other more common conditions, such as tumor ulcer or peptic ulcer disease. Previous articles have solely focused on discussing the high mortality rate, potential causes, and treatment modalities of AEF [10,11,12,13]. There have been no studies exploring the risk factors for AEF in esophageal cancer bleeding patients, with the development of predictive models yet to be explored. To effectively predict the incidence of AEF in this patient population, the development of a risk score is essential. The primary aim of this study is to derive and internally validate a novel prediction score for AEF in patients with esophageal cancer bleeding.

## 2. Materials and Methods

### 2.1. Study Design and Setting

This was a multi-center retrospective observational study. It utilized data from the EDs of five hospitals in Taiwan that shared the same electronic medical records system (EMRs). These hospitals collectively had over 9000 beds and received around 500,000 ED visits annually. All adult patients who met inclusion criteria between 1 January 2010 and 31 May 2022 were included. The study encompassed a dual-stage approach: an initial derivation stage designed to ascertain predictors associated with AEF and construct a novel predictive scoring model, followed by a subsequent validation stage to internally assess and confirm the predictive efficacy of the derived scoring model. This research received ethical approval from the Chang Gung Medical Foundation Institutional Review Board (IRB no. 202301026B0) and adhered to the principles of the Declaration of Helsinki.

### 2.2. Patient Selection and Data Collection

By conducting a search of the EMRs within the study timeframe, we identified all adult patients (≥18 years) diagnosed with esophageal cancer. These patients presented with UGIB caused either by tumor ulcer or AEF during their index ED visits. The identification was based on the following International Classification of Diseases (ICD) codes: ICD-10 codes C15 (Malignant neoplasm of esophagus), K920–K922 (Hematemesis, Melena, Gastrointestinal hemorrhage, unspecified), as well as ICD-9 codes 150.9 (Malignant neoplasm of esophagus, unspecified), 578.9 (Gastrointestinal bleeding), and 578.0–578.1 (Hematemesis, Blood in stool). Patients were excluded if the bleeding etiologies were inconsistent with cancer-associated bleeding, such as gastric/duodenal ulcers, varices, gastritis/duodenitis, or post-operative complications. Patients with missing data for scoring items were subsequently excluded from the model construction and validation phases. The records of the patients selected with EMRs were further reviewed by two physicians for the verification of their inclusion eligibility (S.-W. L. and C.-C. Y.).

Baseline characteristics including sex, age, body mass index (BMI), lifestyle factors (cigarette, betel nut, and alcohol use), initial vital signs and presenting symptoms upon arrival at the ED, medication history, comorbidities (hypertension, diabetes mellitus, coronary artery disease, chronic kidney disease, chronic obstructive pulmonary disease, other malignancy, prior stroke, gastroesophageal reflux disease, and liver cirrhosis), performance status scale by the Eastern Cooperative Oncology Group (ECOG-PS) [14], and Charlson comorbidity index (CCI) [15] were retrieved. The initial symptoms observed at ED included hematemesis, melena, and hematochezia. However, patients with hematochezia were not involved in the study because they were all identified as having lower gastrointestinal bleeding not from tumor ulcer or AEF. Active bleeding was defined as patients with continued bleeding at the ED, such as persistent hematemesis or the passage of melena stool. We collected laboratory data during the initial presentation, such as white blood cell count (WBC), hemoglobin (Hb) levels, platelet count (PLT), prothrombin time (PT), and serum creatinine level. The clinicopathological parameters of the primary cancer were obtained using the latest available data at the time when the bleeding event occurred, including data related to the cancer site, length, tumor-node-metastasis (TNM) stage (based on the TNM staging system by the American Joint Committee on Cancer, 8th edition) [16], cancer treatment modality (including chemoradiation, surgical resection, and stent implantation), and local recurrence.

Emergent endoscopy or computed tomography (CT) angiography was conducted to identify the cause of bleeding and the location of the bleeding site. The therapeutic modalities included supportive care, endoscopic therapy, arterial embolization, esophageal stent implantation, and surgical intervention. Supportive care was defined as the administration of intravenous proton pump inhibitors (PPIs), terlipressin, or tranexamic acid. Endoscopic therapy included methods such as argon coagulation, hemoclip application, epinephrine injection, and band ligation. Moreover, cases necessitating inotropic support, intubation, and subsequent admission to intensive care units (ICUs) were documented. The primary outcome of interest in this study was the diagnosis of AEF, which was confirmed by CT angiography.

### 2.3. Statistical Analysis

#### 2.3.1. Descriptive Comparison and Grouping

Patients in the first stage of the study were randomly assigned to the derivation or validation sets in a 3:2 ratio. This allocation was performed using a straightforward randomization technique with computer-generated random numbers. Patient characteristics, previous medical history, laboratory findings, and presentations of cancer and bleeding were presented as numbers and percentages for the categorical variables, while the continuous data were presented as means ± standard deviations (SDs). Comparisons between the derivation cohort and validation cohort were examined with the Chi-square test or Fisher’s exact test for the categorical variables, and independent Student’s *t*-tests or Mann–Whitney U tests for the normally distributed and skewed continuous variables, respectively. Patients were consistently followed up, with data extending up to the date of their death or their most recent assessment, all of which were extracted from the EMRs. Kaplan–Meier analysis was performed to assess the cumulative survival rate between tumor ulcer bleeding and AEF using the log-rank test. All statistical analyses were performed in SPSS software v26 (SPSS Inc., Chicago, IL, USA) and R Statistical Software (version 3.6.0). A two-sided *p*-value of < 0.05 was considered the threshold of statistical significance. 

#### 2.3.2. Derivation Stage

To identify the predictors of AEF in the derivation cohort, we employed a sequential approach. First, univariable analyses were performed to identify the variables that are potentially associated with AEF. Second, all significant variables from the univariable analyses were inputted into the multivariable logistic regression model to further examine their statistical association. Continuous variables were categorized as a priori using either the standard threshold values established in our laboratory settings or thresholds associated with poor outcomes, as identified in the existing literature [17,18,19,20,21,22,23,24]. In the multivariable model, a *p*-value of < 0.05 was set as the threshold for the incorporation of variables into the final prediction score. The assignment of points to each predictor followed a linear transformation of their respective β-regression coefficients. Each variable’s coefficient was divided by the smallest β-value among the variables included in the final predictive model and then rounded to the nearest integer to determine its contribution to the scoring system. Model discrimination in the prediction score was examined using the c-statistic. Model calibration was assessed by comparing predicted probabilities against observed probabilities and evaluated using the Hosmer–Lemeshow test, where a well-calibrated model is indicated by a *p*-value greater than 0.05. A bootstrapping procedure, involving 1000 samples drawn with replacements from the original cohort, was employed to assess the internal validation of the derived prediction score [25]. According to the occurrence rate of AEF, patients were classified into low- and high-risk categories.

#### 2.3.3. Validation Stage

Validation of the derived prediction score was conducted within an independent validation cohort using the c-statistic. Additionally, the performance of the prediction score was evaluated in terms of its sensitivities, specificities, positive predictive values (PPVs), negative predictive values (NPVs), and weighted accuracy.

## 3. Results

### 3.1. Patient Characteristics of Derivation Cohort and Validation Cohort

A total of 257 esophageal cancer patients satisfied the inclusion criteria of this study. All participants were randomized to a derivation cohort (*n* = 155) and a validation cohort (*n* = 102). AEF occurred in 22 patients (14.2%) in the derivation group and 14 patients (13.7%) in the validation group (Figure 1). No statistically significant difference was observed between these two groups. The patient characteristics in these respective groups, including initial vital signs, presenting symptoms, and laboratory results, were detailed in Table 1. Characteristics related to esophageal cancer, prior cancer treatments, and the various treatment modalities in these two groups were summarized in Table 2. The distributions of age, sex, and initial vital signs (blood pressure, heart rate, and respiratory rate) did not differ between the derivation cohort and validation cohort. The proportions of initial presenting symptoms including hematemesis and melena in both cohorts displayed no significant differences, and similarly, there was no distinction in the proportion of cases with active bleeding between them. Most underlying diseases, including hypertension, diabetes mellitus, coronary artery disease, congestive heart failure, gastroesophageal reflux disease, chronic kidney disease, chronic obstructive pulmonary disease, presence of other malignancy, and liver cirrhosis, had a similar proportion in both groups, except that a history of prior cerebrovascular accident (1.3% vs. 9.8%, *p* = 0.002) was significantly higher in the validation cohort. Initial laboratory parameters were assessed in both cohorts, and none of these parameters exhibited statistically significant differences (Table 1).

The characteristics of esophageal cancer, including length, site, TNM stage, previous cancer treatments, and proportion of local recurrence, did not show any statistically significant differences between the two groups (Table 2). Various treatments were performed for the management of esophageal cancer bleeding, as shown in Table 2, which included supportive care, endoscopic treatment, surgical repair, stent implantation, and arterial embolization. There were also no differences observed in the utilization of these treatments among them. The outcomes, including ICU admission, blood transfusion, intubation, and inotropic agent support, showed no significant difference between the two study cohorts. Kaplan–Meier survival analysis indicated that patients with AEF had a significantly lower survival rate than patients with tumor ulcer bleeding (*p* < 0.001) (Figure 2).

### 3.2. Derivation Stage

Univariable and multivariable logistic regression analyses were used to investigate the predictors of AEF in the derivation cohort (Table 3). The univariable predictors included heart rate > 110 (beats/min) (OR = 2.56; 95% CI 1.02–6.42; *p* = 0.046), respiratory rate > 22 (breaths/min) (OR = 5.17; 95% CI 1.63–16.4; *p* = 0.005), hematemesis (OR = 11.1; 95% CI 1.45–85.2; *p* = 0.021), active bleeding (OR = 13.5; 95% CI 4.73–38.2; *p* < 0.001), previous stent implantation (OR = 5.76; 95% CI 2.01–16.5; *p* = 0.001), Hb < 8 (g/dL) (OR = 3.93; 95% CI 1.54–10.0; *p* = 0.004), PT > 13 (s) (OR = 3.31; 95% CI 1.30–8.39; *p* = 0.012), and serum creatinine level > 1.2 (mg/dL) (OR = 3.95; 95% CI 1.52–10.2; *p* = 0.005). The multivariable analyses indicated that hematemesis (OR = 16.7; 95% CI 1.49–187.1; *p* = 0.022), active bleeding (OR = 5.45; 95% CI 1.37–21.7; *p* = 0.016), previous stent implantation (OR = 8.75; 95% CI 1.78–42.9; *p* = 0.008), PT > 13 (s) (OR = 5.09; 95% CI 1.15–22.6; *p* = 0.032), and Cr > 1.2 (mg/dL) (OR = 9.76; 95% CI 2.37–40.2; *p* = 0.002) were statistically significant predictors of AEF. On the basis of the β-regression coefficient, the coefficient of each variable was divided by 1.627 (the lowest β-regression coefficient value) and rounded to the nearest integer. Then, we established the new prediction score (HEARTS-Score) for AEF (Table 3). In this scoring system, the cumulative score reaches a minimum of 0 points and a maximum of 6 points. Hematemesis is allocated 2 points, while each of the other predictors is assigned 1 point. This score had good discriminative ability, with a c-statistic of 0.90 (95% CI 0.82–0.98) (Figure 3). We performed internal validation through the bootstrapping procedure, yielding a mean c-statistic of 0.90 (95% CI 0.81–0.96). The final predictive model did not exhibit a significant lack of fit, as indicated by the Hosmer–Lemeshow χ² statistic (*p* = 0.254). A calibration plot derived from 1000 bootstrap resamples was displayed in Figure 4. Patients scoring between 0 and 2 points were classified as low risk, with an AEF occurrence rate of 1.2%. Conversely, those with a score greater than 2 were categorized as high risk, and they exhibited a significantly higher AEF occurrence rate of 32.3% (Table 4). Figure 5 illustrates the HEART-S score algorithm, which is employed for predicting AEF.

### 3.3. Validation Stage

Among the 102 patients in the validation cohort, seven had missing data and were excluded, leaving 95 patients for analysis. Using the derived predictive model, 46 patients were identified as low risk, resulting in a 4.3% AEF occurrence rate. Meanwhile, 49 patients were categorized as high risk, demonstrating a higher AEF occurrence rate of 24.5%, as indicated in Table 5. This predictive model effectively distinguished patients at risk, as evidenced by a c-statistic of 0.82 (95% CI 0.72–0.92), shown in Figure 3. Notably, it demonstrated high sensitivity (85.7%) and NPV (95.7%) in high-risk patients. This indicates that individuals classified as low risk exhibited a significantly decreased probability of AEF occurrence, as detailed in Table 6.

## 4. Discussion

This article, to the best of our knowledge, represents the first study to investigate predictors of AEF in patients with esophageal cancer bleeding in the ED. Furthermore, it introduces the HEARTS-Score (Figure 5), the first predictive model designed for AEF detection in this unique population. We performed Kaplan–Meier analysis to assess the cumulative survival rate between tumor ulcer and AEF. The results revealed a significant difference, with AEF demonstrating notably worse long-term and short-term outcomes. Therefore, it is crucial to identify AEF within this patient population. The HEARTS-Score, serving as a predictive model for AEF, consists of five factors, which include hematemesis, active bleeding, PT greater than 13 s, serum creatinine levels exceeding 1.2 mg/dL, and a history of previous stent implantation. This predictive model demonstrates good sensitivity and NPV in high-risk patients. This implies that individuals classified as low risk have a low probability of developing AEF, with rates of 1.2% in the derivation cohort and 4.3% in the validation cohort. Conversely, patients who do develop AEF are highly likely to be categorized as high risk, with a rate of 95.5% in the derivation group and 85.7% in the validation group. Given the critical nature and urgency associated with AEF, we strongly recommend clinicians to prioritize advanced diagnostic imaging, specifically contrast-enhanced chest CT, for high-risk patients to promptly identify AEF. This approach can streamline treatment planning and ultimately minimize the time required to initiate definitive treatment. Additionally, it suggests that hospitals facing resource shortages and unable to perform examinations or treatments should initiate referral procedures as soon as possible. Furthermore, even among patients categorized as low risk, there is still a probability of more than 1% for AEF occurrence in both cohorts. While this might not immediately warrant urgent imaging to exclude AEF, it is imperative to reevaluate the scoring and consider clinical parameters if there are any changes in the patient’s condition. For instance, the emergence of hematemesis, active bleeding, or hemodynamic instability should prompt a reassessment, in conjunction with a clinical evaluation, to judiciously determine the necessity of further diagnostic investigations.

Hematemesis and active bleeding are identified as predictors of AEF in our study. Prior literature has mentioned Chiari’s triad, which consists of mid-chest pain followed by an asymptomatic interval, and subsequently, a sentinel hemorrhage leading to exsanguination within hours to days [10,26]. Based on this triad, it is evident that hematemesis is a common clinical symptom of AEF, regardless of whether it manifests during the sentinel hemorrhage phase or the exsanguination phase. Notably, during the exsanguination phase, patients frequently present with the prominent clinical feature of active bleeding. Previous research consistently reports that more than a third (22 out of 62 or 35.4%) of esophageal cancer patients with UGIB presenting active bleeding were diagnosed with AEF [7]. Therefore, based on this compelling evidence, we consider these two identified predictors to be highly justifiable and clinically significant. While several diseases can present with both hematemesis and active bleeding, such as esophageal varices, gastric varices, and ulcerative diseases, previous studies have discussed the differentiation based on clinical characteristics such as “dark bleeding” and “bright bleeding” [26]. This distinction is grounded in the fact that AEF bleeding primarily stems from arteries, whereas most other causes typically involve venous bleeding. However, due to the clinical differentiation between “bright” and “dark” bleeding entailing a significant degree of uncertainty, we have chosen not to incorporate this distinction within the hematemesis variable. It is important to note that Chiari’s triad is not consistently observed, as one study reported that only 65% of AEF patients exhibited sentinel bleeding [26]. It becomes even more challenging to identify symptom-free intervals effectively in patients who do not exhibit sentinel hematemesis. The use of the HEARTS-Score allows for a more objective assessment of high-risk and low-risk cases, helping to prevent the misdiagnosis of AEF in patients who are in the asymptomatic phase or those who do not exhibit sentinel bleeding or active bleeding. Simultaneously, it enables us to seize the opportunity for timely diagnosis and intervention during symptom-free intervals.

In our study, prior stent implantation has been established as a predictor for the development of AEF. AEF can arise from a multifactorial etiology, with previous research indicating that the most common predisposing factors are thoracic aortic aneurysm (51.2%), followed by foreign body ingestion (18.6%), esophageal malignancy (17%), and surgical complications (4.8%), which include procedures such as esophagectomy and esophageal stent implantation [26]. Patients with advanced esophageal cancer, for whom surgical resection is not a viable option and who present with severe dysphagia, often undergo the insertion of a metal stent to reestablish the patency of the esophageal lumen, as standard medical practice frequently entails. This intervention is employed as one of the customary palliative treatments in this patient cohort [27,28]. In addition, stent placement is occasionally employed to manage esophageal bleeding, secondary to tumor ulcers or acute esophageal varices [29,30]. Siersema et al. reported that among 804 patients who underwent stent implantation, 30 individuals suffered fatal bleeding events, some of which were attributed to the development of AEF [31,32]. Furthermore, Zhan et al. documented a case series illustrating instances of AEF directly related to stent implantation in their research [33]. The exact etiology of AEF associated with stent placement remains unclear; however, it can be linked to various underlying factors. These include the risk of injury primarily due to the repetitive mechanical actions involved in interventional procedures. Improper placement of the stent at an angle relative to the esophageal wall, which can cause friction between the stent and the esophageal wall during vessel pulsations and respiratory movements, results in AEF. Moreover, the high pressure exerted by the stent on the esophageal wall can impede blood supply, leading to localized ischemia, necrosis, or ulceration. Local inflammation and growth of the tumor beneath the stent are also potential contributing factors [34,35]. Hence, it is both reasonable and crucial to recognize previous stent implantation as a reliable predictor of AEF.

In our study, we identified a PT value exceeding 1.3 s and a serum creatinine level greater than 1.2 mg/dL as predictors of AEF. While this may not appear intuitive, we contend that these findings are attributable to the consequences of massive bleeding. PT is a laboratory test used to measure the coagulation time of blood. It is employed to assess the integrity of the extrinsic and common pathways in the coagulation cascade, which are critical in blood clot formation [36,37]. In cases of massive hemorrhage, where a substantial volume of blood is lost, there can be depletion of clotting factors in the bloodstream, including those associated with the extrinsic and common pathways of coagulation. Consequently, PT may be extended due to the reduced availability of these clotting factors required to initiate and support the clotting process. Massive bleeding can also have a profound impact on renal function. Severe and prolonged hemorrhage can lead to hypovolemia, reducing the perfusion of blood to the kidneys. This reduced renal blood flow can result in acute kidney injury or exacerbate pre-existing chronic kidney disease. Furthermore, the breakdown of red blood cells during massive bleeding can release hemoglobin and myoglobin, both of which have the potential to be nephrotoxic, contributing to renal dysfunction [38,39,40].

We recognized several limitations in the current study. Firstly, the study’s retrospective design may have introduced bias due to limitations in the collection of accurate variables. Secondly, while our model included previous treatment with stent implantation as a significant predictor for AEF, other oncological factors such as the T, N, M stages of cancer were not included in our final model. Our analysis suggested a positive correlation of these stages with AEF occurrence, but these did not reach statistical significance, possibly due to the study’s limited sample size. This limitation underscores the need for larger-scale studies to more accurately assess the impact of these oncological factors on AEF risk. Additionally, the model primarily utilizes quickly accessible clinical parameters, prioritizing utility in emergency settings. Consequently, it does not include certain detailed histological factors such as histological grading, lymphovascular invasion, perineural invasion, and margin status, due to their limited availability in acute care contexts. Thirdly, in this study, we only conducted internal validation. Further research with a larger sample size may be necessary to externally validate and refine this predictive model in the future. Fourthly, this study was conducted within a single country, and the study population primarily consisted of individuals with esophageal SCC. Therefore, it may not be appropriate to extrapolate these results to regions where the incidence of adenocarcinoma surpasses that of SCC. Regional variations in cancer types and their characteristics could impact the generalizability of our findings. Fifthly, not all patients underwent contrast-enhanced chest CT, indicating the potential for underdiagnosis in a minority of cases with AEF, thus introducing a source of inherent variability. However, it is important to note that among those patients who were not diagnosed with AEF, there was no evidentiary support for AEF occurrence up to the final follow-up date of this study. Finally, it is essential to acknowledge that the HEARTS-Score was specifically developed for patients experiencing bleeding associated with esophageal cancer. Consequently, it should not be extrapolated to all cases of UGIB, given the significant differences in underlying characteristics and prognosis between esophageal cancer-related bleeding and other causes of UGIB, such as peptic ulcers [7]. This may limit the practical utility of the HEARTS-Score in the EDs.

## 5. Conclusions

This study introduces a novel model, the HEARTS-Score, for predicting AEF in esophageal cancer bleeding patients. The HEARTS-Score comprises five variables: active bleeding, hematemesis, serum creatinine level > 1.2 mg/dL, PT > 1.3 s, and previous stent implantation. Scores on this scale range from 0 to 6, and it exhibits excellent discriminative capabilities. By employing this model, clinicians can more objectively differentiate between high-risk and low-risk patients, facilitating more efficient clinical decision-making, diagnostic planning, and subsequent treatment strategies.

## Figures and Tables

**Figure 1 cancers-16-00804-f001:**
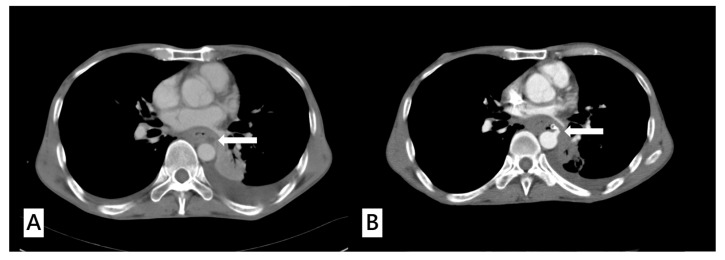
(**A**) A 51-year-old male diagnosed with upper gastrointestinal bleeding caused by advanced thoracic esophageal cancer. The computed tomography angiography (CTA) of aorta revealed a tumor ulcer abutting the aorta with esophageal obstruction (white arrow). (**B**) The same patient presented with hematemesis and was sent to the emergency department 30 days later. Repeated CTA of aorta showed extravasation from the aortic arch abutting the esophagus (white arrow) with a high suspicion of aorto–esophageal fistula.

**Figure 2 cancers-16-00804-f002:**
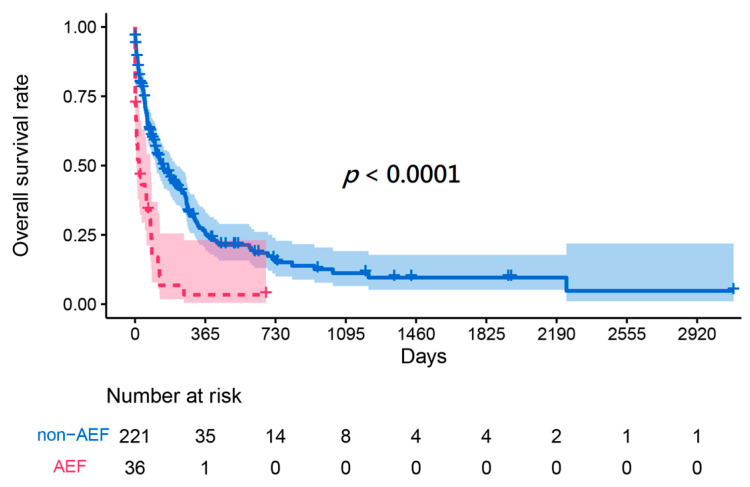
Kaplan–Meier survival curve of esophageal cancer bleeding patients caused by tumor ulcer or arterial–esophageal fistula.

**Figure 3 cancers-16-00804-f003:**
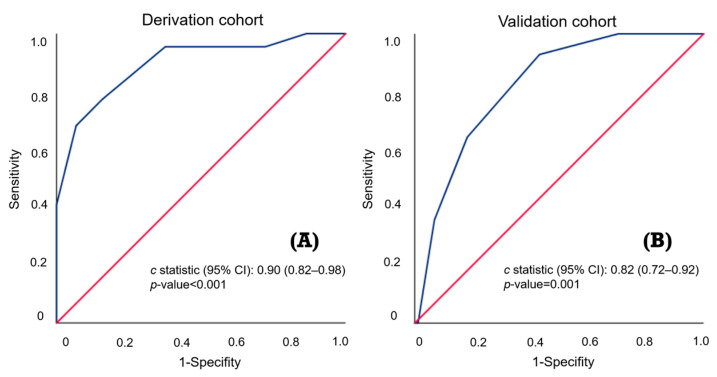
(**A**,**B**) C-statistic of the HEARTS-Score for predicting arterial–esophageal fistula in the derivation cohort (0.90) and validation cohort (0.82). It demonstrates excellent discriminatory ability and achieves statistical significance.

**Figure 4 cancers-16-00804-f004:**
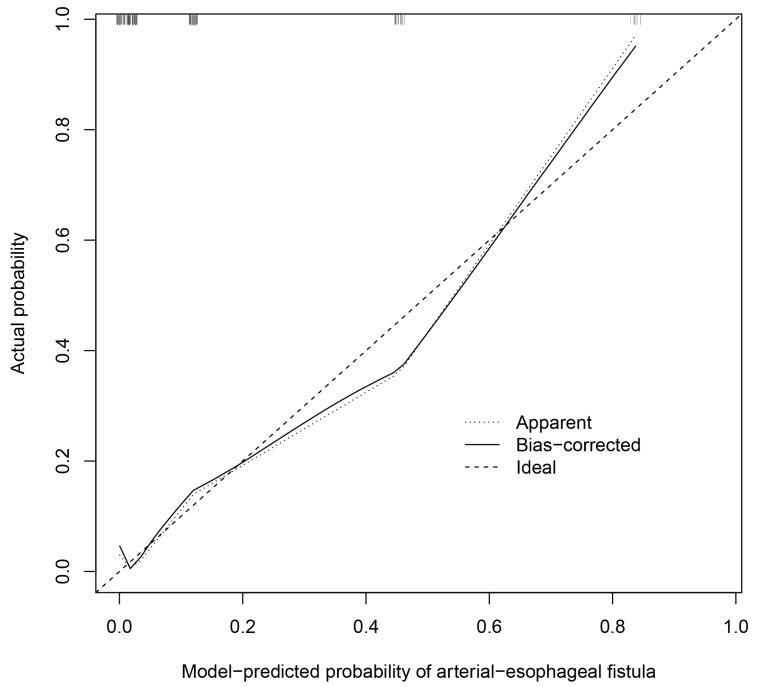
A calibration plot for comparing predicted probabilities against observed probabilities of arterial–esophageal fistula.

**Figure 5 cancers-16-00804-f005:**
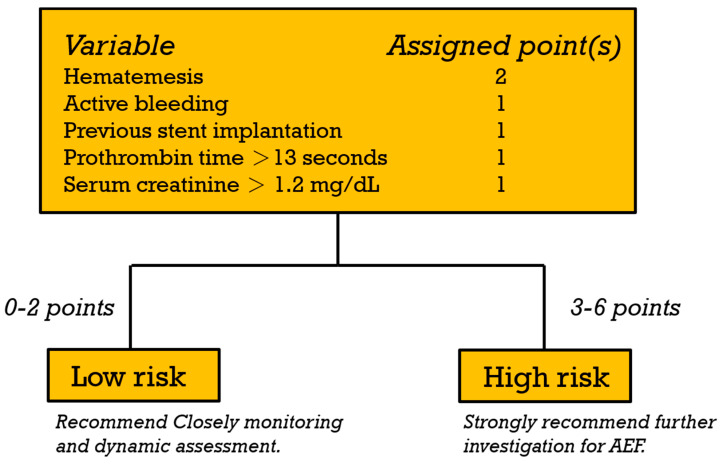
Algorithm of the HEARTS-Score for predicting arterial–esophageal fistula in patients with esophageal cancer bleeding.

**Table 1 cancers-16-00804-t001:** Clinical and demographic characteristics of the derivation and validation cohort.

Variable	Total *n* = 257	Derivation Cohort *n* = 155	Validation Cohort *n* = 102	*p* Value
Age (years)	59.3 ± 11.5	59.2 ± 11.6	59.4 ± 11.3	0.873
Male	249 (96.9)	151 (97.4)	98 (96.1)	0.716
BMI (kg/m^2^)	20.6 ± 3.6	20.6 ± 3.5	20.8 ± 3.8	0.707
ECOG-PS				0.698
0	1 (0.4)	0 (0)	1 (1.0)	
1	188 (73.2)	113 (72.9)	75 (73.5)	
2	46 (17.9)	30 (19.4)	16 (15.7)	
3	18 (7.0)	10 (6.5)	8 (7.8)	
4	4 (1.6)	2 (1.3)	2 (2.0)	
Initial Vital Signs				
SBP (mmHg)	123.4 ± 29.0	120.7 ± 28.5	127.4 ± 29.5	0.072
DBP (mmHg)	76.3 ± 15.9	76.0 ± 16.6	76.7 ± 15.0	0.700
Heart rate (beats/min)	105.6 ± 21.6	104.4 ± 22.1	107.5 ± 20.8	0.266
Respiratory rate (breaths/min)	19.5 ± 2.9	19.6 ± 2.9	19.4 ± 3.0	0.596
Personal Habits				
Smoking history	211 (82.1)	126 (81.3)	85 (83.3)	0.741
Betel nut chewer	130 (50.6)	78 (50.3)	52 (51.0)	1.000
Alcohol consumption	191 (74.3)	116 (74.8)	75 (73.5)	0.884
Comorbidity				
Hypertension	77 (30.0)	42 (27.1)	35 (34.3)	0.266
Diabetes mellitus	39 (15.2)	21 (13.5)	18 (17.6)	0.380
Coronary artery disease	12 (4.7)	7 (4.5)	5 (4.9)	1.000
Congestive heart failure	5 (1.9)	3 (1.9)	2 (2.0)	1.000
Gastroesophageal reflux disease	135 (52.5)	87 (56.1)	48 (47.1)	0.162
Chronic kidney disease	19 (7.4)	11 (7.1)	8 (7.8)	1.000
Prior cerebrovascular accident	12 (4.7)	2 (1.3)	10 (9.8)	0.002*
Liver cirrhosis	44 (17.1)	27 (17.4)	17 (16.7)	1.000
Chronic obstructive pulmonary disease	7 (2.7)	5 (3.2)	2 (2.0)	0.706
Other malignancy	58 (22.6)	33 (21.3)	25 (24.5)	0.648
Charlson comorbidity index	6.92 ± 2.83	6.89 ± 2.84	6.97 ± 2.82	0.824
Current Medication				
Use of NSAIDs	24 (9.3)	12 (7.7)	12 (11.8)	0.381
Use of Anti-platelets Agent^〒^	14 (5.4)	7 (4.5)	7 (6.9)	0.576
Use of Anti-coagulant Agent^†^	1 (0.4)	1 (0.6)	0 (0)	1.000
Use of PPI or H2-receptor antagonist	89 (34.6)	54 (34.8)	35 (34.3)	1.000
Initial Presenting symptoms				
Hematemesis	176 (68.5)	108 (69.7)	68 (66.7)	0.681
Melena	81 (31.5)	47 (30.3)	34 (33.3)	0.681
Active bleeding	64 (24.9)	38 (24.5)	26 (25.5)	0.884
Arterial–esophageal fistula	36 (14.0)	22 (14.2)	14 (13.7)	1.000
Initial Laboratory data				
WBC (10^3^/μL)	11.4 ± 9.5	11.7 ± 10.4	11.2 ± 8.2	0.700
Hb (g/dL)	9.5 ± 2.5	9.6 ± 2.5	9.3 ± 2.4	0.317
PLT (10^3^/μL)	253 ± 138	243 ± 127	269 ± 154	0.140
PT (s)	12.9 ± 6.1	13.1 ± 7.6	12.6 ± 2.7	0.544
Creatinine (mg/dL)	1.05 ± 0.75	1.02 ± 0.53	1.10 ± 1.00	0.364

Count data are expressed as numbers (percentages) and continuous values are expressed as mean ± SD. ECOG-PS: Eastern Cooperative Oncology Group Performance Score; NSAIDs: Non-Steroidal Anti-Inflammatory Drug; BMI: body mass index; PPI: proton pump inhibitor; SBP: systolic blood pressure; DBP: diastolic blood pressure; WBC: white blood cell; Hb: hemoglobin; PLT: platelet count; PT: prothrombin time; ^〒^ Anti-platelet agents including Aspirin (9) or Clopidogrel (2). ^†^ Anti-coagulant agents including Warfarin (1), Apixban (2), Rivaroxaban (3), * *p*-value < 0.05.

**Table 2 cancers-16-00804-t002:** Characteristics of tumor and clinical outcomes of the derivation cohort and validation cohort.

Variable	Total *n* = 257	Derivation Cohort *n* = 155	Validation Cohort *n* = 102	*p* Value
Tumor site (Esophagus)				0.283
Upper third	50 (19.5)	34 (21.9)	16 (15.7)	
Middle third	95 (40.0)	52 (33.5)	43 (42.2)	
Lower third	112 (43.6)	69 (44.5)	43 (42.2)	
Tumor length (cm)	7.11 ± 3.30	6.88 ± 3.05	8.24 ± 4.21	0.119
Tumor pathology				0.283
Squamous cell carcinoma	236 (91.8)	142 (91.6)	94 (92.2)	
Adenocarcinoma	13 (5.0)	8 (5.2)	5 (4.9)	
Small cell carcinoma	1 (0.4)	0 (0)	1 (1.0)	
Melanoma	2 (0.8)	1 (0.6)	1 (1.0)	
Neuroendocrine	1 (0.4)	0 (0)	1 (1.0)	
Unknown	4 (1.6)	4 (2.6)	0 (0)	
T stage				0.378
T1	7 (3.9)	6 (3.9)	1 (1.0)	
T2	25 (9.7)	17 (11)	8 (7.8)	
T3	116 (45.1)	64 (41.3)	52 (51.0)	
T4	105 (40.9)	66 (42.6)	39 (38.2)	
Unknown	4 (1.6)	2 (1.3)	2 (2.0)	
N stage				0.701
N0	40 (15.6)	22 (14.2)	18 (17.6)	
N+	215 (83.7)	132 (85.2)	83 (81.4)	
Unknown	2 (0.8)	1 (0.6)	1 (1.0)	
M stage				0.752
M0	155 (60.3)	90 (58.1)	65 (63.7)	
M1	99 (38.5)	63 (40.6)	36 (35.3)	
Unknown	3 (1.2)	2 (1.3)	1 (1.0)	
Cancer stage				0.728
I	6 (2.3)	5 (3.2)	1 (1.0)	
II	17 (6.6)	10 (6.5)	7 (6.9)	
III	59 (23.0)	34 (21.9)	25 (24.5)	
IV	175 (68.1)	106 (68.4)	69 (67.6)	
Initial cancer treatment				
Surgical resection	17 (6.6)	7 (4.5)	10 (9.8)	0.124
Chemoradiation	170 (66.1)	98 (63.3)	72 (70.6)	0.229
Stent implantation	38 (14.8)	20 (12.9)	18 (17.6)	0.369
Local recurrence	74 (28.8)	45 (29.0)	29 (28.4)	1.000
Emergent examination				
Endoscopy	222 (86.4)	133 (85.8)	89 (87.3)	0.853
Emergent CTA	53 (20.6)	32 (20.6)	21 (20.6)	1.000
Initial medication				
Proton pump inhibitor	235 (91.3)	142 (91.6)	93 (91.2)	1.000
Tranexamic acid	188 (73.2)	116 (74.8)	72 (70.6)	0.474
Terlipressin	32 (12.5)	16 (10.3)	16 (15.7)	0.247
Bleeding treatment				0.687
Endoscopic treatment ^†^	17 (6.6)	10 (6.5)	7 (6.9)	
Surgical repair/Stent implantation	26 (10.1)	13 (8.4)	13 (12.7)	
Arterial embolization	7 (2.7)	4 (2.6)	3 (3.0)	
Blood transfusion	148 (57.6)	86 (55.5)	62 (60.8)	0.440
Intubation	34 (13.2)	22 (14.2)	12 (11.8)	0.707
Inotropic agents support	11 (4.3)	6 (3.9)	5 (4.9)	0.758
ICU admission	29 (11.3)	17 (11.0)	12 (11.8)	1.000
Hospice care	81 (31.5)	49 (31.6)	32 (31.4)	1.000
Rebleeding event	160 (62.3)	92 (59.4)	68 (66.7)	0.293

Count data are expressed as numbers (percentages) and continuous values are expressed as mean ± SD. CTA: computed tomography angiography; ICU: intensive care units. ^†^ Endoscopic treatment included use of Argon coagulation, Hemoclip, Epinephrine injection and Band ligation.

**Table 3 cancers-16-00804-t003:** Univariate and multivariate logistic regression analysis of predictors for arterial–esophageal fistula in the derivation cohort and allocation of points.

	Number of Patients	Univariate		Multivariate		β-Regression Coefficient	Point †
		OR (95%CI)	*p* Value	OR (95%CI)	*p* value		
Age > 65	39	0.60 (0.19, 1.89)	0.382				
Male	151	0.49 (0.48, 4.88)	0.539				
BMI^¶^							
Non-underweight	104	Reference					
Underweight^〒^	41	1.72 (0.65, 4.56)	0.274				
ECOG-PS > 2	11	2.18 (0.54, 8.76)	0.274				
Body temperature > 38℃	6	3.23 (0.55, 18.8)	0.193				
Heart rate > 110 (beats/min)	59	2.56 (1.02, 6.42)	0.046 *	0.94 (0.21, 4.20)	0.933	−0.064	-
RR > 22 (breaths/min)	14	5.17 (1.63, 16.4)	0.005 *	4.18 (0.68, 25.7)	0.123	1.430	-
SBP < 90 (mmHg)	22	1.77 (0.58, 5.35)	0.316				
Underlying disease							
Hypertension	41	0.76 (0.26, 2.22)	0.619				
Diabetes mellitus	20	1.01 (0.27, 3.76)	0.990				
Coronary artery disease	7	1.01 (0.12, 8.80)	0.994				
Congestive heart failure	3	0.00 (0.00, 0.00)	0.999				
Chronic kidney disease	11	0.00 (0.00, 0.00)	0.999				
Prior cerebrovascular accident	2	0.00 (0.00, 0.00)	0.999				
Liver cirrhosis	26	0.72 (0.20, 2.62)	0.615				
Gastroesophageal reflux disease	86	0.49 (0.20, 1.22)	0.125				
Initial presentation							
Hematemesis	108	11.1 (1.45, 85.2)	0.021 *	16.7 (1.49, 187.1)	0.022*	2.816	2
Active bleeding	38	13.5 (4.73, 38.2)	<0.001 *	5.45 (1.37, 21.7)	0.016*	1.695	1
Anti-platelets agents use	7	2.56 (0.47, 14.1)	0.280				
Anti-coagulant agents use	1	0.00 (0.00, 0.00)	1.000				
Tumor location							
Upper third	34	1.41 (0.50, 3.93)	0.515				
Middle third	52	1.81 (0.72, 4.51)	0.206				
Lower third	69	0.42 (0.15, 1.13)	0.086				
Cancer stage^¶^							
T stage > 2	130	1.07 (0.29, 3.98)	0.918				
N stage > 0	132	1.68 (0.36, 7.79)	0.506				
M stage > 0	63	1.36 (0.54, 3.41)	0.519				
Cancer treatment							
Surgical resection	7	0.01 (0.12, 8.80)	0.994				
Chemoradiation	98	2.98 (0.96, 9.30)	0.060				
Stent implantation	20	5.76 (2.01, 16.5)	0.001 *	8.75 (1.78, 42.9)	0.008 *	2.169	1
Local recurrence	45	2.33 (0.93, 5.88)	0.072				
Initial Laboratory data							
WBC > 11.0 (10^3^/μL)	62	1.61 (0.65, 3.98)	0.304				
Hb < 8.0 (g/dL)	38	3.93 (1.54, 10.0)	0.004 *	2.76 (0.60, 12.7)	0.195	1.013	-
PLT < 150 (10^3^/μL)	37	0.93 (0.317, 2.72)	0.892				
PT > 13 (s)^¶^	51	3.31 (1.30, 8.39)	0.012 *	5.09 (1.15, 22.6)	0.032 *	1.627	1
Cr > 1.2 (mg/dL)^¶^	33	3.95 (1.52, 10.2)	0.005 *	9.76 (2.37, 40.2)	0.002 *	2.278	1

OR: odd ratio; 95% CI: 95% confidence interval; ECOG-PS: Eastern Cooperative Oncology Group Performance Score; BMI: body mass index; RR: respiratory rate; SBP: systolic blood pressure; WBC: white blood cell; Hb: hemoglobin; PLT: platelet count; PT: prothrombin time; Cr: creatinine. * *p*-value < 0.05. ^〒^ Underweight: BMI < 18.5 kg/m_2_. ^†^ The assignment of points to predicting AEF was based on a linear transformation of the corresponding b-regression coefficient. The coefficient of each variable was divided by 1.627 (the lowest β-Regression coefficient value) and rounded to the nearest integer. ^¶^ There were missing data (13 cases in BMI, 15 cases in PT, and 3 cases in Cr, 2 cases in T stage, 1 case in N stage, 2 cases in M stage).

**Table 4 cancers-16-00804-t004:** Distribution of risk scores and risk classification in a prediction score (HEARTS-Score) for arterial–esophageal fistula in the derivation cohort.

Risk Score (6-Point Scoring System)				Risk Classification			
Total Points	Patients (*n* = 146) *	AEF	Rate of AEF, %	Risk Category	Patients	AEF	Rate of AEF, % (95% CI)
0	15	0	0.0	Low risk	81	1	1.2 (0–6.7)
1	19	1	5.3				
2	47	0	0.0				
3	36	5	13.9	High risk	65	21	32.3 (21.2–45.0)
4	20	7	35.0				
5	9	9	100.0				
6	0	0	NA				

AEF: arterial–esophageal fistula; NA: not applicable; 95% CI: 95% confidence interval; HEARTS-Score consists of hematemesis, active bleeding, renal function (serum creatinine level > 1.2 mg/dL), PT > 13 s, previous stent implantation; low risk is defined as a HEARTS-Score of 0–2 points, while high risk is defined as a HEARTS-Score of 3–6 points. * There were missing data in nine cases.

**Table 5 cancers-16-00804-t005:** Distribution of risk scores and risk classification in a prediction score (HEARTS-Score) for arterial–esophageal fistula in the validation cohort.

Risk Score (6-Point Scoring System)				Risk Classification			
Total Points	Patients (*n* = 95) *	AEF	Rate of AEF (%)	Risk Category	Patients	AEF	Rate of AEF, % (95% CI)
0	10	0	0.0	Low risk	46	2	4.3(0.5–14.8)
1	13	0	0.0				
2	23	2	8.7				
3	35	6	17.1	High risk	49	12	24.5(13.3–38.9)
4	11	5	45.5				
5	2	1	50.0				
6	1	0	0.0				

AEF: arterial–esophageal fistula; 95% CI: 95% confidence interval; HEARTS-score consists of hematemesis, active bleeding, renal function (serum creatinine level > 1.2 mg/dL), prothrombin time > 13 s, previous stent implantation; low risk is defined as a HEARTS-Score of 0–2 points, while high risk is defined as a HEARTS-Score of 3–6 points. * There were missing data in seven cases.

**Table 6 cancers-16-00804-t006:** Diagnostic ability of the HEARTS-Score in the derivation cohort and validation cohort.

Derivation Cohort			Validation Cohort		
C-Statistic (95% CI): 0.90 (0.82–0.98)			C-Statistic (95% CI): 0.82 (0.72–0.92)		
	Low risk	High risk		Low risk	High risk
Patients, *n* (%)	81(55.5)	65(44.5)	Patients, *n* (%)	46(48.4)	49(51.6)
AEF, *n* (%)	1(1.2)	21(32.3)	AEF, *n* (%)	2(4.3)	12(24.5)
Sensitivity, %	4.5	95.5	Sensitivity, %	14.3	85.7
Specificity, %	35.5	64.5	Specificity, %	45.7	54.3
PPV, %	1.2	32.3	PPV, %	4.3	24.5
NPV, %	67.7	98.8	NPV, %	75.5	95.7
Weighted accuracy, %	20.0	80.0	Weighted accuracy, %	30.0	70.0

AEF: arterial–esophageal fistula; PPV: positive predictive value; NPV: negative predictive value; 95% CI: 95% confidence interval; HEARTS-Score consists of hematemesis, active bleeding, renal function (serum creatinine level > 1.2 mg/dL), prothrombin time > 13 s, previous stent implantation; low risk is defined as a HEARTS-Score of 0–2 points, while high risk is defined as a HEARTS-Score of 3–6 points.

## Data Availability

The authors confirmed that the data supporting the findings of this study are available from the electronic medical records system in Chang-Gung Hospital, which is available from the corresponding author upon reasonable request.

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
