# Peer review of "Novel Prediction Score for Arterial–Esophageal Fistula in Patients with Esophageal Cancer Bleeding: A Multicenter Study"

_cancers, 2024, doi:10.3390/cancers16040804_

Round 1

Reviewer 1 Report

Comments and Suggestions for Authors

Excellent study, well conducted and extensively discussed.

I have no special concerns. Well done.

can patients at risk be added to the K-M curves?

Comments on the Quality of English Language

Minor polishing is required.

Author Response

Reviewer 1

Excellent study, well conducted and extensively discussed.

I have no special concerns. Well done.

can patients at risk be added to the K-M curves?

Response:

Thank you for your feedback. We have revised our Kaplan-Meier curves as follows:

Revised K-M curve

Reviewer 2 Report

Comments and Suggestions for Authors

Manuscript entitled "A novel prediction score for arterial-esophageal fistula in patients with esophageal cancer bleeding: A Multicenter Study"

Major issues:

1.  The authors should include histological factor of esophageal cancer including but not limited to histological type, differentiation, 

2. The authors should include the site specific factors of esophagus cancer (as in cancer resistry) into analysis.  

Comments on the Quality of English Language

Accetable

Author Response

Reviewer 2

Comment

The authors should include histological factor of esophageal cancer including but not limited to histological type, differentiation

The authors should include the site specific factors of esophagus cancer (as in cancer resistry) into analysis. 

Response:

Thank you for your valuable suggestions! We appreciate your insights and agree that histological factors play a significant role in understanding esophageal cancer.

Our prediction model is designed with the primary aim of utilizing quickly available parameters that can be easily accessed by frontline physicians in emergency settings. In this regard, we have included common oncological factors such as TNM stage, recurrence status, cancer site, and pathology, which are generally available and relevant in acute scenarios.

However, other factors like lymphovascular invasion, perineural invasion, and margin status, while undoubtedly important, are often not immediately available in the context of emergency care. The inclusion of such detailed histological information would potentially limit the practical utility of our model in urgent clinical situations.

Furthermore, our current database does not contain some of these detailed histological factors. Accessing and integrating this information would require a considerable amount of time, which could be impractical for the intended use of our model in emergency settings.

We acknowledge this as a limitation of our study and believe it could be an area for future research. If more comprehensive data become available, it would certainly be worthwhile to explore the impact of these additional histological factors on our prediction model.

We are grateful for your constructive feedback and have mentioned these considerations and limitations in our revised manuscript to provide a clearer understanding of our study's scope and objectives.

Discussion, section 5, line 106-109

Additionally, the model primarily utilizes quickly accessible clinical parameters, prioritizing utility in emergency settings. Consequently, it does not include certain detailed histological factors such as histological grading, lymphovascular invasion, perineural invasion, and margin status, due to their limited availability in acute care contexts. Thirdly, this study was…

Reviewer 3 Report

Comments and Suggestions for Authors

Comments to the Author

This multicenter, retrospective study examined the development of aorto-esophageal fistulas (AEF) in esophageal cancer bleeding. Using logistic regression analysis, a predictive model of AEF was constructed and verified in a validation cohort, concluding that the HEART score is valuable in AEF diagnosis.

The manuscript is well written and has a clear message. However, I also have some comments.

Comment 1

Oncological factors may play a significant role in AEF in esophageal cancer. That is, AEF is most likely to form in the advanced esophageal cancer. However, oncologic factors were not risk factors for AEF in this study. Additional discussion of this should be added.

Comment 2

Adding censoring and the number at risk to the survival curves in Figure 2.

Comment 3

The introduction section needs a stronger statement on what makes this manuscript special compared to prior literature.

Comment 4

The limitations section needs to be more developed.

Author Response

Reviewer 3

This multicenter, retrospective study examined the development of aorto-esophageal fistulas (AEF) in esophageal cancer bleeding. Using logistic regression analysis, a predictive model of AEF was constructed and verified in a validation cohort, concluding that the HEART score is valuable in AEF diagnosis.

The manuscript is well written and has a clear message. However, I also have some comments.

Comment 1

Oncological factors may play a significant role in AEF in esophageal cancer. That is, AEF is most likely to form in the advanced esophageal cancer. However, oncologic factors were not risk factors for AEF in this study. Additional discussion of this should be added.

Response:

Thank you for your insightful comment! In our current model, we included previous treatment with stent implantation as a significant predictor for AEF. However, we acknowledge that other oncological factors, such as the cancer stage, were not included in the final model. Our analysis did indicate a positive correlation of T, N, M stages with the occurrence of AEF, but these did not reach statistical significance. We hypothesize that this might be due to the relatively small sample size of our study, which may have limited the statistical power necessary to detect the effect of these stages on AEF occurrence conclusively.

We recognize this as a limitation of our current research and agree that a study with a larger sample size would be beneficial to investigate these relationships more comprehensively. Future research in this area could provide valuable insights into how different oncological factors, including the stages of cancer, contribute to the risk of developing AEF in patients with esophageal cancer.

We have included these considerations and potential avenues for future research in the limitation section of our revised manuscript.

Discussion, section 5, line 100-109

We recognized several limitations of the current study. Firstly, the study's retrospective design may have introduced bias due to limitations in the collection of accurate variables. Secondly, while our model included previous treatment with stent implan-tation as a significant predictor for AEF, other oncological factors such as the T, N, M stages of cancer were not included in our final model. Our analysis suggested a positive correlation of these stages with AEF occurrence, but these did not reach statistical sig-nificance, possibly due to the study's limited sample size. This limitation underscores the need for larger-scale studies to more accurately assess the impact of these oncolog-ical factors on AEF risk. Additionally, the model primarily utilizes quickly accessible clinical parameters, prioritizing utility in emergency settings. Consequently, it does not include certain detailed histological factors such as histological grading, lymphovascu-lar invasion, perineural invasion, and margin status, due to their limited availability in acute care contexts. Thirdly, in this study, we only conducted…

Comment 2

Adding censoring and the number at risk to the survival curves in Figure 2.

Response:

Thank you for your valuable feedback. We have revised our Kaplan-Meier curves as follows:

Revised K-M curve

Comment 3

The introduction section needs a stronger statement on what makes this manuscript special compared to prior literature.

Response:

Thank you for your suggestion! We have added a description in the introduction to express the uniqueness of this article as follows:

Introduction, section 2, line 72-75

However, due to its episodic and nonspecific symptoms, AEF is often initially misdi-agnosed as other more common conditions, such as tumor ulcer or peptic ulcer disease. Previous articles have solely focused on discussing the high mortality rate, potential causes, and treatment modalities of AEF[10-13]. There have been no studies exploring the risk factors for AEF in esophageal cancer bleeding patients, with the development of predictive models yet to be explored. To effectively predict…

Comment 4

The limitations section needs to be more developed.

Response:

Thank you for your comments ! We have strengthened the limitations section as follows:

Discussion, section 5, line 100-109

We recognized several limitations of the current study. Firstly, the study's retro-spective design may have introduced bias due to limitations in the collection of accurate variables. Secondly, while our model included previous treatment with stent implan-tation as a significant predictor for AEF, other oncological factors such as the T, N, M stages of cancer were not included in our final model. Our analysis suggested a positive correlation of these stages with AEF occurrence, but these did not reach statistical sig-nificance, possibly due to the study's limited sample size. This limitation underscores the need for larger-scale studies to more accurately assess the impact of these oncolog-ical factors on AEF risk. Additionally, the model primarily utilizes quickly accessible clinical parameters, prioritizing utility in emergency settings. Consequently, it does not include certain detailed histological factors such as histological grading, lymphovascu-lar invasion, perineural invasion, and margin status, due to their limited availability in acute care contexts. Thirdly, in this study, we only conducted internal validation. Further research with a larger sample size may be necessary to externally validate and refine this predictive model in the future.

Discussion, section 5, line 120-126

However, it is important to note that among those patients who were not diagnosed with AEF, there was no evidentiary support for AEF occurrence up to the final follow-up date of this study. Finally, it's crucial to recognize that the pre-dictive model was built on the assumption that the bleeding originates from tumor ulcers. Therefore, it is applicable exclusively to individuals experiencing bleeding associ-ated with esophageal cancer and cannot be extrapolated to all cases of GIB in patients with esophageal cancer. The substantial differences in the underlying characteristics between these two groups result in significant variations in mortality rates. This may limit practical utility of this scoring system in ED.

Reviewer 4 Report

Comments and Suggestions for Authors

Thank you for this well written and performed study. 

Some comments:

1. The study population only includes esophageal cancer associated bleeding - hence the scoring system may not be applicable to all esophageal cancer patients who present with upper GI bleeding, limiting the practical utility of this scoring system in real life. 

2. The definition of active bleeding is vague. "Active bleeding was defined as patients with continued bleeding at the ED." Please elaborate what is the definition of "continued bleeding at the ED", as clinically, it is often quite subjective. 

3. The authors nicely show that the derivation and validation cohort are quite similar in variables/parameters. While this lends strength to internal validity, the score still has to be validated externally for it to be useful.

4. line 270 states that " those categorized as low-risk had a significantly reduced likelihood of developing AEF". The scoring system does not predicct for the development of AEF. Rather, it tries to predict which patient already has an AEF at presentation. 

5. The authors state in the discussion that the score is dynamic in nature, and that it can be repeated if there are any changes in the patient's condition. Could the authors show data to support that the score has been validated for dynamic use?

6. The main benefit of using the score seems to be to prioritize contrast enhanced CT imaging, useful in a resource limited country. 

Comments on the Quality of English Language

Acceptable

Author Response

Reviewer 4

Comment 1

The study population only includes esophageal cancer associated bleeding - hence the scoring system may not be applicable to all esophageal cancer patients who present with upper GI bleeding, limiting the practical utility of this scoring system in real life.

Response:

Thank you for sharing your thoughts! We have added a paragraph in the limitations section as follows:

Discussion, section 5, line 120-126

there was no evidentiary support for AEF occurrence up to the final follow-up date of this study. Finally, it is essential to acknowledge that the HEARTS-Score was specifically developed for patients experiencing bleeding associated with esophageal cancer. Conse-quently, it should not be extrapolated to all cases of UGIB, given the significant differ-ences in underlying characteristics and prognosis between esophageal cancer-related bleeding and other causes of UGIB, such as peptic ulcers[7]. This may limit the practical utility of the HEARTS-Score in the EDs.

Comment 2

The definition of active bleeding is vague. "Active bleeding was defined as patients with continued bleeding at the ED." Please elaborate what is the definition of "continued bleeding at the ED", as clinically, it is often quite subjective.

Response:

Thanks for your insight ! We have provided a clearer definition of active bleeding as follows:

Materials and Methods, section 2.2. Patient Selection and Data Collection, line 114-116

having lower gastrointestinal bleeding not from tumor ulcer or AEF. Active bleeding was defined as patients with continued bleeding at the ED, such as persistent hematemesis or the passage of melena stool. We collected laboratory data during…

Comment 3

The authors nicely show that the derivation and validation cohort are quite similar in variables/parameters. While this lends strength to internal validity, the score still has to be validated externally for it to be useful.

Response:

Thank you very much for your feedback! We have revised the limitations section as follows:

Discussion, section 5, line 110-112

Consequently, it does not include certain detailed histological factors such as histolog-ical grading, lymphovascular invasion, perineural invasion, and margin status, due to their limited availability in acute care contexts. Thirdly, in this study, we only con-ducted internal validation. Further research with a larger sample size may be necessary to externally validate and refine this predictive model in the future. Fourthly, this study was…

Comment 4

line 270 states that " those categorized as low-risk had a significantly reduced likelihood of developing AEF". The scoring system does not predicct for the development of AEF. Rather, it tries to predict which patient already has an AEF at presentation. 

Response:

Thank you for your suggestion ! We have revised the sentence as follows:

Results, section 3.2. Validation stage, line 273-275

in high-risk patients. This indicates that individuals classified as low-risk exhibited a significantly decreased probability of AEF occurrence, as detailed in Table 6.

Comment 5

The authors state in the discussion that the score is dynamic in nature, and that it can be repeated if there are any changes in the patient's condition. Could the authors show data to support that the score has been validated for dynamic use?

Response:

Thank you very much for your concise feedback! We indeed lack evidence to substantiate the dynamic nature of this score. As a result, we have revised the discussion section accordingly:

Discussion, section 1, line 24-30

Additionally, it suggests that hospitals facing resource shortages and unable to perform examinations or treatments should initiate referral procedures as soon as possible. Furthermore, it's worth noting that the HEARTS-Score is dynamic in nature. even among patients categorized as low-risk, there is still a probability of more than 1% for AEF occurrence in both cohorts. While this might not immediately warrant urgent im-aging to exclude AEF, it is imperative to reevaluate the scoring and consider clinical parameters if there are any changes in the patient's condition. For instance, the emer-gence of hematemesis, active bleeding, or hemodynamic instability should prompt a reassessment, in conjunction with a clinical evaluation, to judiciously determine the necessity of further diagnostic investigations.

Comment 6

The main benefit of using the score seems to be to prioritize contrast enhanced CT imaging, useful in a resource limited country. 

Response:

Thank you very much for your comments. We have added a paragraph to the discussion section as follows:

Discussion, section 1, line 22-24

This approach can streamline treatment planning and ultimately minimize the time required to initiate definitive treatment. Additionally, it suggests that hospitals facing resource shortages and unable to perform examinations or treatments should initiate referral procedures as soon as possible. Furthermore, even among patients categorized as low-risk…

Round 2

Reviewer 2 Report

Comments and Suggestions for Authors

The revision is accepted for publication.

Comments on the Quality of English Language

Acceptable

Reviewer 3 Report

Comments and Suggestions for Authors

This article has already well-reviewed and able to provide very important information. I recommend that this manuscript is accepted for publication in Cancers.

Reviewer 4 Report

Comments and Suggestions for Authors

Thank you for addressing my comments adequately.

Comments on the Quality of English Language

acceptable